# *Solanum tuberosum* Microtuber Development under Darkness Unveiled through RNAseq Transcriptomic Analysis

**DOI:** 10.3390/ijms232213835

**Published:** 2022-11-10

**Authors:** Eliana Valencia-Lozano, Lisset Herrera-Isidrón, Jorge Abraham Flores-López, Osiel Salvador Recoder-Meléndez, Aarón Barraza, José Luis Cabrera-Ponce

**Affiliations:** 1Departamento de Ingeniería Genética, Centro de Investigación y de Estudios Avanzados del IPN, Unidad Irapuato, Irapuato 36824, Guanajuato, Mexico; 2Unidad Profesional Interdisciplinaria de Ingeniería Campus Guanajuato (UPIIG), Instituto Politécnico Nacional, Av. Mineral de Valenciana 200, Puerto Interior, Silao de la Victoria 36275, Guanajuato, Mexico; 3CONACYT-Centro de Investigaciones Biológicas del Noreste, SC. IPN 195, Playa Palo de Santa Rita Sur, La Paz 23096, Baja California Sur, Mexico

**Keywords:** transcriptome-wide analysis, microtubers, potato, *Solanum tuberosum*, darkness, cell cycle, ribosomal proteins, PEBP family genes, cytokinin

## Abstract

Potato microtuber (MT) development through in vitro techniques are ideal propagules for producing high quality potato plants. MT formation is influenced by several factors, i.e., photoperiod, sucrose, hormones, and osmotic stress. We have previously developed a protocol of MT induction in medium with sucrose (8% *w*/*v*), gelrite (6g/L), and 2iP as cytokinin under darkness. To understand the molecular mechanisms involved, we performed a transcriptome-wide analysis. Here we show that 1715 up- and 1624 down-regulated genes were involved in this biological process. Through the protein–protein interaction (PPI) network analyses performed in the STRING database (v11.5), we found 299 genes tightly associated in 14 clusters. Two major clusters of up-regulated proteins fundamental for life growth and development were found: 29 ribosomal proteins (RPs) interacting with 6 PEBP family members and 117 cell cycle (CC) proteins. The PPI network of up-regulated transcription factors (TFs) revealed that at least six TFs–MYB43, TSF, bZIP27, bZIP43, HAT4 and WOX9–may be involved during MTs development. The PPI network of down-regulated genes revealed a cluster of 83 proteins involved in light and photosynthesis, 110 in response to hormone, 74 in hormone mediate signaling pathway and 22 related to aging.

## 1. Introduction

Potato (*Solanum tuberosum* L.) is the most important staple food worldwide just after maize, rice, and wheat with an approximate production of around 360 million tons in 2020 on a surface area of 16.5 million hectares [1]. Potato is a geophyte plant with a dual reproduction system, forming vegetative storage organs called tubers and a sexual propagation that is tightly regulated to ensure fitness in adverse climatic conditions [2].

The potato tubers are swollen underground storage organs differentiated from the subapical meristem of the stolon (modified stem) mediated by radial divisions, expanding to further differentiate into starch accumulating parenchyma. Tubers function in vegetative propagation (eyes; axillary meristems) and as storage enabling survival during winter months and/or prolonged periods of abiotic stress (perennialism) and are a means to bypass sexual reproduction.

The molecular mechanisms of the potato tuberization process have been well documented and depend on light (day length and light quality), temperature, and nutrient supply [3,4,5].

Plant biotechnology techniques have been applied to produce potato tubers in vitro called microtubers (MTs). The MTs process occurs when axillary buds are cultured in medium containing high content of sucrose, hormones, and different light quality/darkness. MTs have advantages in terms of storage, transport, and cultivation, due to their reduced size and weight, requiring less space compared to seedlings, with a higher multiplication rate producing seed potato faster and cheaper than other methods [6,7,8,9].

We developed a protocol for MTs induction in potato *S. tuberosum* var Alpha culturing stolon explants in MS medium supplemented with 8% SUC, high content of gelrite 6.0 g/L, and cytokinin (CK) 2iP 10 mg/L under darkness. The rationale of this protocol was that two-component and cytokinin (CK) signaling interacts with homeobox TFs; ribosomal proteins (RPs), cell cycle (CC), carbon metabolism, auxin responsive factors, and stem cell maintenance genes were involved in this process [10].

Several transcriptomic analyses during potato tuberization have revealed conserved molecular machinery during this process: PEBP members (florigen and tuberigen genes), photoperiod, transcriptional regulation, protein synthesis, cell cycle regulation, metabolic pathways, secondary metabolism, starch and sucrose metabolism, hormone signaling, among others [11,12,13,14,15,16]. 

Moreover, a few papers have published reliable gene ID in order to compare results. Sharma et al. [15] published the targets of StBEL5 in tuber development by a transcriptome analysis. A set of more than 2000 up-regulated genes can be read from the Appendix A. A PPI network in STRING data base v11.5 with the highest confidence (0.900) revealed that the up-regulated genes are involved in fundamental processes for life: RPs, CC, spliceosome, ribonucleoprotein complex, and basal TFs interacting with several clusters of genes involved in light, stress and metabolic pathways. 

Light modulates the molecular mechanisms during potato tuberization (photomorphogenesis); little attention has been given to the molecular events in the dark. To elucidate the molecular mechanisms of MTs development under these conditions, we performed a transcriptomic-wide analysis.

Differential expressed genes (DEG) obtained from the analysis were screened in a bioinformatic tool STRING database v11.5 (https://string-db.org/cgi/input?sessionId=bbmDlM5pze5B&input_page_active_form=multiple_identifiers) with the highest confidence (0.860). 

The PPI network of up-regulated TFs revealed that at least six TFs–MYB43, TSF, bZIP27, bZIP43, HAT4, and WOX9–may be involved during MTs development. These TFs were involved in vascularization, proliferation process, cell fate commitment, epigenetic regulation, meristem organization, cell cycle, and ribosomes. 

The PPI network of down-regulated TFs demonstrated several TCP TFs families known as repressor of shoot morphogenesis interacting with MYB TFs involved in flower development.

The PPI network of up-regulated genes revealed two clusters–one made of RPs, that play an important role in the control of cell growth, division, and development [17].

PEBP family members, StSP6A, StSP3D, StSP5G, CEN1/BFT, ATC, and FD TF interact with RPL11. The second cluster belongs to CC, whose function is to correctly duplicate the DNA present in the genome and to segregate the resulting copies correctly into the two daughter cells [18]. The CC cluster interacts tightly with several metabolic pathways: one and carbon metabolism, CK response, citrate cycle, fatty acid metabolism, thylakoid, sulfur metabolism, disulfide isomerase, oxidative stress, and amino sugar and nucleotide sugar metabolism. Down-regulated genes were associated with photosynthesis, light harvesting complex, response to light, hormone signal transduction, response to hormone, flowering, and plant organ development.

In this manuscript we have made a deep analysis by text mining, using the STRING data base from which we proposed a new model of MTs development under darkness conditions.

## 2. Results

### 2.1. MTs Transcriptomic-Wide Analysis 

In order to identify the genes involved in the development of MTs under dark conditions, a transcriptomic-wide analysis was performed. We sequenced cDNA libraries constructed from two treatments: MR8-G6-2iP inducing MTs and MR1-G3-2iP, and non-inducing MTs using the Illumina HiSeq 2000 platform. This produced a total of 397,834,274 reads, from all six cDNA libraries from MR8-G6-2iP and MR1-G3-2iP. A summary of mapping statistics obtained for each sample is given in Table 1.

Overall, 4839 DEG were found in MR8G62iP (MTs inducing treatment) versus control line (non-MTs); 2896 were up-regulated and 1843 down-regulated.

Gene Ontology (GO) enrichment analysis of up-regulated genes identified 2713 genes with annotated GO terms. 

GO biological process includes 440 significantly enriched GO terms. Both up- and down-regulated biological process were related to protein phosphorylation, regulation of transcription, transmembrane transport, carbohydrate metabolic process, and proteolysis. 

Specific up-regulated biological processes include: microtubule-based movement, translation, fatty acid biosynthetic process, lipid transport, DNA replication, gycolitic process, positive regulation of transcription, cell wall modification, DNA repair, defense response, carboxylic acid metabolic process, protein dephosphorylation, gluthatione metabolic process, and the amino acid metabolic process. Specific down-regulated biological processes include: photosynthesis, light harvesting, cell wall biogenesis, cellular glucan, and the xyloglucan metabolic process (Figure 1).

GO cellular components include 82 significantly enriched GO-terms; 64 were up-regulated and those specific of MTs were: ribosome, nucleosome, microtubule, extracellular region, and cytoplasm. Specific down-regulated cellular components include: cell wall, extrinsic component of membrane, photosystem I and II (Figure 2).

GO molecular function includes 132 significantly enriched GO-terms; the seven main functional subgroups are: GO:0003824 catalytic activity (963), GO:0005488 binding (998), GO:00116491 oxidoreductase activity (243), GO:0043167 ion binding (604), GO:0016787 hydrolase activity, GO:0098772 molecular function regulator, and GO:0008092 cytoskeletal protein binding (67).

### 2.2. TFs

Overall, 112 up-regulated and 145 down-regulated TFs were found in the transcriptomic-wide analysis of MTs induced under dark conditions. 

#### 2.2.1. Up-Regulated TFs

Up-regulated TFs are related to C2H2-type zinc finger (20), F-box (15), bHLH (13), AP2 ERF (13), MYB (10), GATA (6), ZF-HD protein (3), Basic leucine zipper (5), bZiP (5), B3 DNA binding d2ggomain (5), NF-Y (4), WRKY (3), TCP (2). 

The PPI network of up-regulated TFs made on the STRING data base v11.5 with medium confidence revealed that 23 out of 112 TFs interact (Figure 3), while 14 interact with at least 1 TF. The list of up-regulated TFs interacting in the PPI network and levels of gene regulation are shown in the heat-map (Figure 4).

#### 2.2.2. Down-Regulated TFs

Overall, 145 TFs were found to be down-regulated in MTs development. These TFs are related to bHLH (38), Homeobox (30), AP2/ERF (21), SANT/Myb domain (19), Myb domain (19), Zinc finger C2H2 superfamily (18), F-box (14), B-box-type zinc finger (11), WRKY (9), CCT domain (8), B3 DNA binding domain (7), Transcription factor TCP (6), BTB/POZ domain (6), and SKP1/BTB/POZ (6).

The PPI network of down-regulated TFs revealed that 58 out of 112 TFs interact (Figure 5), while 9 interact with at least 1 TF. 

The list of down-regulated TFs interacting in the network and levels of gene regulation are shown in the heat-map (Figure 4).

#### 2.2.3. PPI Network of Up- and Down-Regulated TFs

The PPI network of up- and down-regulated TFs revealed both sets of TFs found interact in separated clusters–one forming MTs and the another corresponding to down-regulation TFs involved in shoot-stolon differentiation, gametophytes, plant organ development, and hormone signaling (Appendix A).

### 2.3. Up-Regulated Genes

From 1699 up-regulated genes, 299 were tightly associated in two main clusters (Figure 6). Two fundamental processes for life were detected: RPs (29) and CC (117), interacting directly with proteins that sense the environment: osmotic stress (21), oxidative stress (9), CK response (23), one carbon metabolism (6), carbon metabolism (38), TCA cycle (16), acyl carrier proteins (6), fatty acid metabolism (14), thylakoid (13), redoxins (9), sulfur metabolism (8), disulfide isomerase activity (5), and immunophilins (12). Three clusters comprising 37 genes do not interact: sterol and terpenoid biosynthesis (16) and amino sugar and nucleotide sugar metabolism (7) (Figure 6).

### 2.4. Down-Regulated Genes

Overall, 271 out of 1616 down-regulated genes were present in the PPI network with high confidence (0.700). Since MTs were induced in darkness, the network consists in one cluster related to photosynthesis (69) tightly interacting with several functions (Figure 7). These functions include; light harvesting complex (26), chloroplast (134), biosynthesis of secondary metabolites (65), carbohydrate derivative binding (27), cellular lipid metabolic process (17), starch and sucrose metabolism and glycosyl hydrolases (14), carbohydrate metabolic process (38), carbon metabolism (28), cellular lipid metabolic process (17), homeobox and calmodulin binding motif (2), response to hormone (11), flowering (1), plant hormone signal transduction (2), and response to blue light (10) (Figure 7).

### 2.5. DEG Related to PEBP Family Members

Overall, six PEBP family members were present in MTs development, three FT orthologs of PEBP family are present in potato, StSP6A, StSP5G (PGSC0003DMT400041726), and StSP3D (PGSC0003DMT400041725), two CEN1-BFT (PGSC0003DMT400030575), CEN1-BFT (PGSC0003DMT400037143), and a self-pruning ATC (PGSC0003DMT400018307). The TF FD (PGSC0003DMT400061403) a bZIP transcription factor-27 self-pruning was found interacting with ATC (Figure 8). 

Gene ontology (GO) from functional annotation revealed a relationship between florigen genes present in the PPI network and RPs. bZIP27 FD (PGSC0003DMT400061403) was associated in 35 out of 59 GO-terms in which RPs were present (GO:0009889, GO:0009891, GO:0009893, GO:0010468, GO:0010556, GO:0010557, GO:0010604, GO:0010628, GO:0019222, GO:0031323, GO:0031325, GO:0031326, GO:0031328, GO:0048522, GO:0050789, GO:0050794, GO:0051171, GO:0051173, GO:0060255, GO:0065007, GO:0065009, GO:0080090, GO:2000112, GO:0005488, GO:0097159, GO:0098772, GO:0140110, GO:1901363, GO:0005622, GO:0005634, GO:0043226, GO:0043227, GO:0043229, GO:0043231, GO:0110165). 

In the case of StSP6A, it was associated in 12 out of 51 GO-terms in which RPs were present (GO:0009987, GO:0050789, GO:0065007, GO:0005488, GO:0005622, GO:0005634, GO:0005737, GO:0043226, GO:0043227, GO:0043229, GO:0043231, GO:0110165). 

StSP3D (PGSC0003DMT400041725 and PGSC0003DMT40004172a) were associated in 12 out of 51 GO-terms in which RPs were present (GO:0009987, GO:0050789, GO:0065007, GO:0005488, GO:0005622, GO:0005634, GO:0005737, GO:0043226, GO:0043227, GO:0043229, GO:0043231, GO:0110165). 

*CEN1/BFT/TFL1C* genes (PGSC0003DMT400030575 and PGSC0003DMT400037143) were associated in 3 out of 17 GO-terms in which RPs were present (GO:0005622, GO:0005737, GO:0110165). The list of PEBP family members and levels of gene regulation are shown in the heat map (Figure 9).

### 2.6. DEG Related to RPs 

The PPI network analysis for the up-regulated transcripts corresponding to RPs in the established MTs developmental conditions yielded a cluster of 31 RPs interacting proteins, from which 11 proteins corresponded for the small subunit and 18 for the large subunit, three for eukaryotic elongation factors, two for WD40, RACK1, and ATARCA, and two for glutathione S-transferase genes (Figure 10). The architecture of the RPs cluster yielded in this analysis is quite similar to the evolutionary traits previously determined in PPI network analyses present in the Eukarya domain [19]. Moreover, in the PPI network analysis we were able to disentangle the protein interactions in four main categories. The RPs transcripts that were activated under high sucrose content conditions were highlighted in yellow. Those RPs transcripts that were activated under high sucrose content/darkness conditions, according to Gamm et al. [20], were highlighted in green. The RPs transcripts that were activated under high sucrose content/light conditions were highlighted in blue [20] as well as the RPs transcripts that were activated under high sucrose content/CKs according to Brenner and Schmülling [21]. We also found that among the RPs transcripts was present RPL29e (PGSC0003DMT400069847) that directly interacts with FKBP12 (PGSC0003DMT400083778) and RPL11 (PGSC0003DMT400031869), which in turn interacts with florigens and the thioredoxin-dependent protein (PGSC0003DMT400035271), the ubiquitin fusion protein, and the proliferating cell nuclear antigen (PGSC0003DMT400078207).

### 2.7. DEG Related to CC

We found 117 up-regulated transcript genes directly related to the CC of potato MTs under darkness condition, from which 36 transcripts were directly related with mitotic regulation, 31 transcripts corresponded to E2F factors, 25 transcripts were related with cytoskeleton polymerization, 5 transcripts were related with motor proteins, 13 with epigenetic regulation through DNA/Histone methylation, and 5 with histone/nucleosome binding (Figure 11 and Appendix A).

#### 2.7.1. Cyclins/CDKs

Two CDKs: CDC2-CDKA (PGSC0003DMT400075349) and CDC20-CDKA (PGSC0003DMT400071215) were found among the up-regulated transcripts in this work. The plant CDC2 is homologous to CDKA and in mammals to CDK1 and CDK2. CDKA function has been characterized as a regulatory protein at the G1/S and G2/M checkpoints. In addition, among the up-regulated transcripts related to Cyclins, we found 10 cyclins, which correspond to cyclin-A (four transcripts), cyclin-B (three transcripts), and cyclin-D (three transcripts). Furthermore, in the PPI network analysis the CDKA (PGSC0003DMT400071215) protein interacts with all the cyclins A and B in the network (Figure 11 and Appendix A). The list of CC genes and levels of gene regulation are shown in the heat map (Figure 12).

#### 2.7.2. E2F Factors

In this work, we found up-regulated transcripts belonging to the E2F TFs family, which are directly involved in the regulation of the DNA replication in the onset of S phase. Among those transcripts, the PCNA2 (PGSC0003DMT400078207) interacts with E2F3 (PGSC0003DMT400042347), which in turn interacts with *THY-1* (PGSC0003DMT400001937). The THY-1 interacts with LOG10, PGSC0003DMT400057413) which their transcriptional activation is CK-dependent. Further, PCNA2 interacts with SWI/SNF2 (PGSC0003DMT400034983), RNR1 (PGSC0003DMT400060658), and TSO2 (PGSC0003DMT400052217). The E2F3 transcription factor also interacts with BRCA1 (PGSC0003DMT400023253) (Appendix A).

#### 2.7.3. Histone Binding Proteins

Among the proteins present in the CC cluster, we also found three histone proteins: H2A (PGS003DMT40009751), H2A.1 (PGS0003DMT40003677), and H2AXB (PGSC0003DMT00034958). Those three proteins interact with the plant protein BRCA1 that in turn interacts with ORTHRUS 2 (PGSC000DMT400084297) and PCNA2 (PGSC0003DMT400078207). Moreover, ASF1 (PGSC0003DMT400078369) interacts with PCNA2 and with THY-1.

#### 2.7.4. PPI Network Interaction of CC with RPs and Florigens

The PPI network analyses shed light on to protein interactions and allow a glimpse as to their biological and functional relationship. In this regard, we found that PEBP family members including the tuberigen StSP6A interact with RPs via RPL11 (PGSC0003DMT400031869), then with RPL29e (GSC0003DMT400069847) and UBQ1 (PGSC0003DMT400000364) within the cluster of RPs. UBQ1 interacts with the first protein of CC, PCNA2 (PGSC0003DMT400078207) (Figure 11). PCNA2 interacts with CDT1A (PGSC0003DMT00028257). CDT1A interacts with CDC6 (PGSC0003DMT400078033), then with CDC2a (PGSC0003DMT400075349), followed with CDC45 (PGSC000EDMT400071831) and with B1-type cyclin-dependent kinase (PGSC0003DM7400071215) interacting with CYCLIN B1 (PSC0003DMT400015245) and with B2-type cyclin-dependent kinase (PGSC0003DMT400025910) (Figure 11). In addition, CDT1A interacts with Exonuclease 1 (PGSC0003DMT400018710), with HUS2 (PGSC0003DMT400007294), then with BRCT domain BRCA1 (PGSC0003DMT400023253), with BRCA1 (PGSC0003DMT400013749), with Ubiquitin-conjugating enzyme e2-16 (PGSC0003DMT400009620), MCM4 (PGSC0003DMT400079535), then with MCM2 (PGSC0003DMT400005594), and with Timeless-interacting (PGSC0003DMT400080556) and two replication proteins RPA2 (PGSC0003DMT400035809) and RPA70B (PGSC0003DMT400063329).

### 2.8. DEG Transcripts with Biological Functions 

The list of the transcripts genes found in the DEG analysis under MTs developmental darkness condition were directly related to cytokinin response, osmotic stress, one-carbon, carbon metabolism, glycolysis/gluconeogenesis, citrate cycle, fatty acid metabolism, thylakoids, acyl-carrier proteins, redoxins, response to oxidative stress, immunophilins, ribosomal proteins, sulfur metabolism, mitotic cell cycle, E2F activation, cytoskeleton, motor proteins, methylation, histone binding, sterol biosynthetic process and terpenoid backbone biosynthesis, amino sugar and nucleotide sugar metabolism, disulfide isomerase activity, LRR; leucine rich repeats, and sucrose metabolism, which are available in Appendix A.

### 2.9. Validation of the Transcriptome-Wide Analysis 

Validation of the transcriptomic-wide analysis was made by selecting 19 DEG and analyzing their regulation by quantitative reverse transcription PCR (qRT-PCR), using the primers described in Appendix A. The results are shown in Figure 13, indicating that the values are consistent with those obtained in the transcriptomic-wide analysis.

The genes selected to validate de transcriptomic-wide analysis of MTs development were: StSP6, ATC, UBQ1, FKBP12, RPL29, RPL11, PRXQ, MDH1, SHM1, METK2, TPI, TPP, SPP, BRCA1, RNR1, TSO2, THY-1, LOG10, SWI/SNF, PCNA2, and E2F3. EFa1, SEC3 were used as endogens.

## 3. Discussion

MTs development were induced from a stolon in medium containing sucrose 8%, gelrite 6 g/L, a CK 2iP 10 mg/L under darkness. In this environment, light modulating/repressing signals from leaves to underground were absent.

The molecular mechanisms of tuber development depend on genes related to light perception, temperature, and nutrient supply. These mechanisms have been associated mainly with phytochromes (PHYB and PHYF), homeobox TFs (BEL5 and POTH1), florigens belonging to the PEBP gene family (FT, StSP3D, StSP6A, StSP5G, TFL1), St14-3-3, StFDL1, TAC complex, StABI5 (StABL1), and several molecular regulators controlled under length day conditions (for review see [3,4,5]). 

### 3.1. PPI Network of Up-Regulated TFs

TFs are proteins that control the transcription of genes by binding to a defined region of the genome. TFs are essential for the regulation of biological process, such as body planning, development, differentiation, and responses to various environmental signals [22]. 

MTs development showed 23 up-regulated TFs interacting in a PPI network (Figure 3). The interpretation of the molecular mechanisms from a stolon to a tuber development in high content of sucrose and gelrite, 2iP as CK under darkness is as follows: 

The MYB44 TF (PGSC000DMT400008569) a tuber-specific and sucrose-responsive element binding factor. MYB44 confers resistance to abiotic stresses dependent on ABA. Enrichment analysis with 50 interactors revealed that MYB44 is involved in auxin signaling pathways and tissue vascularization. MYB44 interacts with an AFR protein (PGSC0003DMT400004486), F-box/kelch-repeat skip25-like. AFR protein interacts with another tuber-specific and sucrose-responsive element binding factor TSF (PGSC0003DMT000794169). Enrichment analysis with 50 interactors revealed that TSF is involved in cell fate specification and commitment, multicellular organism development, cell cycle regulation, spliceosomal and ribonucleoprotein complex. TSF interacts with bZIP43 (PGSC0003DMT400003516); it is expressed in inflorescence meristem during petal differentiation and expansion stage. An enrichment analysis with 50 interactors revealed that bZIP43 is involved in histone modification, h3-k9 methylation, ribonucleoprotein complex, 14-3-3 interaction, and amino acid biosynthesis. bZIP43 interacts with a bZIP27-like, self-pruning TF (PGSC0003DMT400061403), required for positive regulation of flowering. Enrichment analysis with 50 interactors revealed that bZIP27 is involved in structural component of ribosomes, cell cycle regulation, cyrcadian rhythm, protein transport, methylation. bZIP43 interacts with a wuschel-related homeobox 9-like (WOX9), COMPOUND INFLORESCENCE (PGSC0003DMT400027850). WOX9 is required for meristem growth and development, and acts through a positive regulation of wuschel. Mutant phenotypes include embryo lethality, reduced root development, and smaller meristems. Phenotypes can be rescued by the addition of sucrose in the growth media. Overexpression can partially rescue the triple mutants of CK receptor phenotype, as a downstream effector of CK signaling. Enrichment analysis with 50 interactors revealed that WOX9 is involved in ribosome, cell cycle regulation, cell fate commitment, meristem growth and maintenance, translation initiation factor 4f, and flower development.

bZIP43 also interacts with a homeobox-leucine zipper protein HAT4 (PGSC0003DMT400067540). HAT4 acts as mediator of the red/far-red light effects on leaf cell expansion in the shading response. It is also involved in the negative regulation of cell elongation and proliferation processes, such as root formation and secondary growth of the vascular system. HAT4 TF interacts with 6 bhlh TFs. Enrichment analysis with 50 interactors revealed that HAT4 is involved in flower development, meristem maintenance, photoreception, sugar signaling, and abiotic stimulus.

The above information about PPI network of up-regulated TFs during MTs development may explain partially some molecular events required to develop MTs: vascularization (MYB44), proliferation process (HAT4), cell cycle, ribosomes (bZIP27), cell fate commitment and meristems (WOX9), light regulation (HAT4), and epigenetic regulation (bZIP43) present during MTs differentiation. The rest of TFs in the PPI network may participate during MTs development (Figure 3). Further analysis of each component is required to elucidate it.

### 3.2. PPI Network of Down-Regulated TFs

MTs development revealed that 57 down-regulated TFs interact in a PPI network (Figure 5). The morphological transformation from a stolon explant (modified stem) to a tuber must involve the down-regulation of TFs involved in stem, leaf, root, and flower development. From 57 down-regulated TFs, 14 TFs were related to shoot system development, 7 to flower and gametophyte development, 6 to phyllome development, 32 in response to hormone, and 23 in hormone signaling pathways.

The PPI network revealed a cluster of 18 homedomain TF superfamily/16 MYB interacting with two BTI TF (PGSC0003DMT400039384) and (PGSC0003DMT400069392) involved in gametophyte development. The MYB TF are involved in multicellular organism development, stamen maturation and dehiscence, seed coat, lateral organ boundaries, and root and xylem differentiation (Figure 5). The gametophyte cluster interacts with a light regulated plant development consisting in BBX17, PIL6 (phytochrome interacting factor), two FAMA TFs, involved in differentiation of stomatal guard cells, derepresses stem cell gene expression, and 10 TF required during fertilization of ovules by pollen.

Another cluster interacting with the gametophyte is composed by TCP TFs. These TFs are involved in leaf development, branching, circadian rhythm, embryonic growth, floral organ morphogenesis and hormone signaling. Recently, Nicolas et al. [23], demonstrated that the TCP TF BRC1b is a tuber repressor in aerial axillary buds, limits sucrose accumulation, promotes dormancy, abscisic acid response, and a reduced number of plasmodesmata. In addition, it interacts with StSP6A, and inhibits its tuber-inducing activity in aerial nodes. In our work, StBRC1b (PGSC0003DMT400010380) was found to be down-regulated and interacting within the network (Figure 5). StBRC1b interacts with TCP23-like and F-box/lrr-repeat MAX2 homolog a-like. A combined up- and down-regulated TFs networks revealed that StBRC1b interacts close to one up-regulated TF, the tuber-specific and sucrose-responsive element TSF (PGSC0003DMT000794169). STBRC1b interacts with MAX2 (PGSC0003DMT400020318) and KUF1 (PGSC0003DMT400004486), a karrikin protein that interacts with TSF TF, up-regulated in this work (Appendix A).

Tang et al. [24], reported that the gene Soltu.DM.06GO25210, named by authors IT1, is identical to StBRC1b. When silenced by CRISPR/Cas, stolons were converted to branches instead of swelling with the capability of developing small tubers within time. StSP6A interacts with IT1 (StBRC1b), and authors suggest that it forms a complex during tuber initiation. They also found that there was an impaired interaction between the StSP6A gene from Etuberosum, a non-tuber producer Solanum plant species, due to the deleted fourth exon in Etuberosum. MAX2 is required for responses to strigolactones and karrikins as well as the repression of axillary shoots outgrowth. MEE1 is required during endosperm development in embryos.

### 3.3. Up-Regulated DEG PPI Network

#### 3.3.1. PEBP Family Members and FD TF Interacting with RPs

The PEBP gene family is well-conserved across species from animals to plants, which play central roles in several biological processes. In *Homo sapiens*, *PEBP1 (RKIP)* has been implicated in several cancers, probably acting as a metastasis suppressor [25]. In addition, it has been proposed to be an active regulator for growth, transformation, differentiation [26], and a cell cycle modulator [27]. In flowering plants, PEBP gene family members have undergone duplication. PEBP are involved in determination of plant architecture. PEBP family consists of three groups: the *FLOWERING LOCUS* (*FT*)-like, the *TERMINAL FLOWER1* (*TFL1*)-like, and *CENTRORADIALS* (*CEN*)-like. The *MOTHER OF FT AND TFL1*-like (*MFT*), and the *BROTHER OF FT AND TFL1* (*BFT*)-like are present as a separate subclade within the CEN group [28]. In potato, multiple *FT* paralogues are related to both sexual and vegetative reproduction. SP3D, an FT orthologue, controls flowering, while SP5G and SP6A, two other PEBP genes, act as a repressor and inducer of tuber formation, respectively [29]. The balance between *FT*/*TFL1* defines the plant growth habit as indeterminate or determinate by modulating the pattern of formation of vegetative and reproductive structures in the apical and axillary meristems [30]. In our study, six PEBP family members were transcriptionally active in MTs development in darkness conditions, such as three orthologs *SP6A*, *StSP5G* (PGSC0003DMT400041726), *SP3D* (PGSC0003DMT400041725), and two orthologs of *CEN1*-*BFT* (PGSC0003DMT400030575). In the PPI network analysis, we found that all PEBP protein family members interact with the universal plastid RPL11 (PGSC0003DMT400031869). PEBPs lack a DNA-binding domain but can exert the function as transcriptional co-regulators through complexing with TFs, regulating the expression of downstream target genes [31]. The TF FD (PGSC0003DMT400061403), a bZIP transcription factor-27 self-pruning was also found interacting with ATC (Figure 8).

Purwestri et al. [32] and Wang et al. [33], through a yeast two-hybrid screening approach, found that SP6A can interact with RPs and other proteins involved in protein synthesis, RNA and DNA binding proteins, histones, initiation factors, signaling and carbon metabolism, and regulate flowering. In addition, the PPI network analysis unveiled 77 interactors, widely distributed throughout the cells in nearly all subcellular components, including plastids, ribosomes, thylakoids, cytosol, and chloroplasts. SP6A functions as positive regulator of tuber development under high contents of sucrose [30,34] and represses bud flower formation under short day conditions [35]. It is involved in several pathways during tuberization, including temperature, microRNA, and photoperiod [36,37,38]. Several transcriptomic analysis of potato *S. tuberosum* have revealed the presence of RPs during the tuberization process [12,15,16,39,40]. In Sharma et al. [15], we found through PPI network analysis with the highest confidence (0.900) that RPs proteins interacted with SP6A, RP40SA, RP60S, RPS8, RPS4A, RPL10, BEL5, and RPL14. 

The ABI5-like 1 (StABL1) bind to SP6A in a 14-3-3 manner forming an alternative tuberigen activation complex (aTAC), thereby resulting in early tuberization [41]. In our study, the transcript for abscisic acid-insensitive 5-like protein (PGSC0003DMT400072260) was up-regulated; it did not interact directly with SP6A in the PPI network analysis, even using the lower confidence (0.150) value. This suggest that a 14-3-3 protein is essential to establish a protein interacting complex with SP6A. In our study, the 14-3-3, (34G; PGSC0003DMT400079172) transcript was up-regulated, and its protein interacts with SP6A, using a high confidence (0.780) value. It is very likely that SP6A and 14-3-3 might be forming a complex under MTs darkness conditions. 34G (14-3-3) is involved in cell cycle regulation, mitotic cycle, ribosome biogenesis, response to stress, starch and sucrose metabolism, reproductive process, and basal transcription factor [42,43].

*CENTRORADIALS* (*CEN*) represses the initiation of floral meristems in several species. The *DHvCEN5* barley plants developed spikelet initiation in both photoperiods; GO enrichment analysis demonstrate that *HvCEN* had a negative impact on to chromatin modification, cell cycle regulation, CK signaling, and cell growth through RPs [44]. A member of the *TERMINAL FLOWER-1/CENTRORADIALS* gene family (*CEN*) has shown to function as a negative regulator of tuberization. At the protein level, CEN competes with SP6A for the interaction with other members of the TAC complex [45]. Moreover, in potato plants, *CEN* overexpression induced a lower rate of sprout growth, lower content of CKs, and higher content of abscisic acid (ABA) [46].

Another transcript for a *FT* was found in this work–the *SP5G* (PGSC0003DMT400041726) which was up-regulated. The function of this *FT* is to blockade the tuberization process. In the absence of light/darkness, SP5G might not repress *SP6A*. In addition, it has been shown that PHYB and PHYF co-stabilized CONSTANS-like 1 protein (COL1), binds SP5G, and represses *SP6A* and tuberization [47]. We also found that the transcript for *SP3D* was up-regulated and that this protein is involved in flowering, and it has been shown that florigens can promote tuberization; therefore, the SP3D might be able to promote this process [29,48].

#### 3.3.2. RPs Cluster

Several transcriptomic analyses in *S. tuberosum* have revealed the transcriptional activation of the RPs coding genes in the tuberization process [12,15,16,20,39,40]. Specifically, the work of Taylor et al. [49] found a 15-20-fold increase in transcript levels of two RPs, S19 and L7, during the early stages of tuberization in stolon tips. Moreover, the RP proteins have been directly related with growth, development, embryo viability, flowering, and leaf development [50,51,52,53,54,55,56].

##### Sucrose Activation of RPs

The up-regulation of RPs in Arabidopsis is activated by different biotic and abiotic factors [57,58,59,60,61,62,63] such as sucrose [20,64,65,66,67] and CKs [21].

In our study, a high content of sucrose (8%), a CK 2iP 10 mg/L, and osmotic stress conditions were applied under darkness conditions. Hummel et al. [67] found that in response to 6% sucrose feeding, 166 out of 204 RPs were up-regulated. In our study, 31 RPs were detected, and according to the data obtained by Gamm et al. [20] who applied 5% of sucrose, we obtained 14 RPs activated in dark conditions and 12 with similar in light and dark RPs activated by sucrose.

##### CK Activation of RPs

In our study, 12 transcripts for RPs were directly related to CKs activation [21], 12 transcripts for RPs were transcriptionally activated by both sucrose and CK treatment. In *Arabidopsis,* 87 transcripts for RP of the 474 identified responded specifically to CKs; CK-responsive RPs related genes encode proteins for both plastidic and cytoplasmatic localization [21].

CKs binds directly to ribosomes in higher plants with high affinity [68,69]. CKs alter the polyribosome formation, ribosomal protein synthesis, number of ribosomes, and the phosphorylation status of ribosomal proteins [70,71,72,73,74,75,76,77]. In *Arabidopsis*, Karunadasa et al. [74] demonstrated that protein synthesis is induced by CKs and CK signaling and requires isoforms of the RP L4. The loss of function of RPL4 increases the tolerance to osmotic stress and decreases sensitivity to CK-induced growth. In our study, 12 RPs were activated by CKs [21], and 12 RPs were activated by both sucrose and CKs treatment. In *Arabidopsis*, 87 out of 474 respond to CKs [21]. The CK-responsive genes encode proteins of both plastidic and cytoplasmatic localization.

##### Stress Activation of RPs

The expression levels of RPs are altered under stress conditions [57,58,61,62,63,75], suggesting that it participates in signal detection and stress response. In rice, RPs large subunits are water and salt stress responsive proteins [61,62,63]. In maize, ribosome footprints from chloroplast and cytosol have been analyzed [76] and provided insights into the dynamic translation mechanisms during drought [77] and viral infection [78]. In cotton, RPL14-2 interact with several RPs to activate drought and salt tolerance [79]. In tea (*Camellia sinensis* L.), RPL32 is up-regulated in drought stress [80].

##### RPs and Their Relationship with CC

In *Saccharomyces cerevisiae,* the repression of nine RPs 60S genes (L4, L7, L18, L3, L17, L28, L35, L37, and L40) resulted in G2/M phase arrest. The repression of nine other 60S RPs (L1, L3, L9, L16, L19, L21, L25, L30, and L43) and 22 40S genes (S0, S1, S2, S3, S4, S5, S6, S9, S10, S13, S14, S15, S17, S19, S20, S21, S22, S26, S27, S29, and S30) caused arrest in the G1 phase [81].

Overall, 14 of 38 proteins analyzed in yeast were present in the potato RPs network: RPL1, RPL18, RPL25, RPL35, RPL40, RPS2, RPS3, RPS6, RPS9, RPS13, RPS15, RPS17, RPS29, and RPS30. Accordingly, they could contribute to the activation of CC during potato MTs development.

Maitra et al. [82] demonstrated a relationship between RPs, CC, and one-carbon metabolism. Yeasts regulate the one-carbon flow balance between donors by controlling the activity of cytoplasmic serine hydroxymethyl transferase (SHM2P) [83]. In cells lacking certain ribosomal proteins, SHM2 translation efficiency and Gly:Ser ratios were found to be low. These observations are consistent with low flow through the 1C pathways. 

We proposed that RPs present in our analysis, in addition to being essential in the progression of CC in MTs development, could also participate in the one-carbon pathways that ultimately comprise a series of interrelated metabolic pathways that include the methionine and folate cycles–critical for cell function–providing 1C units (methyl groups) for the synthesis of DNA, polyamines, amino acids, creatine, and phospholipids [84].

#### 3.3.3. CC Cluster

Tuber development relies on the temporal and spatial control of cell proliferation and cell growth to correctly develop an organ that will help the potato plant to survive under adverse conditions. Potato tubers develop from stolons, from which swelling growth occurs in the subapical region of the meristem. Initial expansion and radial cell division of cells in the pith and cortex, in combination with restricted longitudinal growth is followed by cell division and enlargement in random orientations in the perimedullary region once the swelling has reached a diameter of 2 to 4 mm, continuing until the tuber reaches its full size. In this study, genes involved in CC regulation were present, cyclins/CDKs, E2F factors and histone binding proteins.

##### Cyclins/CDKs

Overall, 2 cyclin-dependent kinase (CDKs); CDC2 (PGSC0003DMT400075349) and CDKB1;2 (PGSC0003DMT400071215), 10 cyclins: 4 cyclin-A, 3 cyclins-B and 3 cyclin-D were found. CDKB1;2 (PGSC0003DMT400071215) interacts with cyclins A and B present in our network. CKs in the presence of sucrose activate the expression of cyclins-D that have been shown to be essential in CC regulation. Yang et al. [85] demonstrated that CKs (Zeatin, BAP, and 2iP) regulate cell cycle by promoting the nuclear shuttling of the MYB-domain protein 3R4. MYB3R4 binds 5′-AACGG-3′ motifs of promoters of G2/M-specific genes and E2F targets. In our analysis, a gene homologous of MYB3R4 was found (PGSC0003DMT00047825). In accordance, it is very feasible that the potato MYB3R4 activate the cell cycle regulation genes during MTs development.

Rhee et al. [86] analyzed cyclins-D gene expression in potato exposed to different light regimes, hormones (CKs and auxins), and carbohydrates (SUC and glucose) and found that *StCYCD3;1* and StCYCD3;3 were more expressed in above-ground organs whereas StCYCD3;2 was expressed more abundant in undergrounds organs. The above expression levels were when cytokinins like zeatin and BAP combined with sucrose induced higher expression compared to 2,4-D. In our work, StCYCD3;2 was found interacting within the PPI network with a cyclin-dependent kinase B2 (CDKB) (PGSC0003DMT400025910). The levels of expression found were similar; *StCYCD3;2* was 3.5, while *StCYCD3;1* was 3.03, and *StCYCD3;3* was a 2.85 fold change.

In *H. sapiens*, *PEBP1* (*RKIP*) is a metastasis suppressor in aggressive cancers [25] and cell cycle modulator [27]. Silenced-*RKIP* accelerate DNA synthesis and G1/S transition entry by inducing the expression of *CDC6*, *MCM2,4,6,7*, *cyclinD2*, *D1*, and *E2*. It also enhances G2/M transition and down-regulates G2/M checkpoints like *AURORA-B*, cyclin G1, and slow G2/M transition time. This led to a higher proliferation rate compared to control cells [27].

In our work, the components of G2/M checkpoints were present, showing similarities with the function of PEBP1(RKIP) in *H. sapiens*.

The CC regulation of mitosis is summarized as follows: The CYCB-CDKA complex (PGSC0003DMT400071215) interacts with MAD2 (PGSC0003DMT400045025) (mitotic arrest deficient), MAD2 interacts with CDC20 protein (PGSC0003DMT400047585) (APC metaphase/anaphase complex). MAD2 and CDC20 are checkpoint proteins locally catalyzed by kinetochores, the central player in the distribution of genetic material to the daughter cells during mitosis [87].

MAD2 interacts with EB1c (PGSC0003DMT400064236) (microtubule stability). MAD2 interacts with TUB1 (PGSC0003DMT400037093) (tubulin formation). In Arabidopsis, *TUB1* expression is linked to light induction [88]. 

In hypocotyl explants, the *TUB1* highest expression levels are found in dark conditions. The *TUB1* transcriptional behavior in previous reports is in accordance with our data, since our work was carried out in dark conditions [88]. 

MAD2 interacts with TUB8 (PGSC0003DMT400028800). During the initiation of tuber formation, there are changes in the expression of tubulins at the tip of the stolons [87].

MAD2 interacts with AURORA2 (PGSC0003DMT400065470), a serine/threonine protein kinase. It is abundant in tissues with high division, such as roots and flowers, but low in leaves and stems. In mitosis it is intimately linked with the cytoskeleton (pre-prophase band, phragmoplast, metaphase plate) required for cytokinesis [89].

MAD2 interacts with KRP-130 (PGSC0003DMT400078142). KRP-130 are involved in microtubule dynamics, morphogenesis, chromosome segregation, organelle transport, and vesicles [90].

KRP-130 interacts with BUB1 (PGSC0003DMT400082809); BUB1 binds to the kinetochore and has an important role in establishing the mitotic spindle checkpoint and chromosome aggregation [91]. 

BUB1 interacts with the CDKA-CYCA complex (PGSC0003DMT400071215). BUB1 interacts in the same way with the complex of CDKA-CYCA. CDKA (PGSC0003DMT400071215) interacts with four B cyclins. CDKA has been identified at the G1/S and G2/M checkpoints [92] and is modulated by cyclins A and B [93].

##### E2F Factors 

The E2F proteins form a family of TFs that regulate the transition from the G_1_ to the S phase in the cell cycle. We found 31 E2F factors in this analysis. The activation of genes such as *PCNA2*, *RNR1*, *TSO2*, *THY-1*, *SWI/SNF2*, *ORC6*, *ORC4*, *ORC5*, *SPT16*, *MCM6*, *SMC3*, *MCM2*, *CHR11*, *POLA2*, and *PSF2* demonstrates that DNA replication was carried out correctly, validating that the conditions evaluated are optimal for the development of MTs. Abnormal induction of the S phase re-entry in mature leaf cells is observed in overexpression of E2F in Arabidopsis [94]. Unregulated activity of this family of TFs has been characterized broadly in human cancers [95].

Abiotic stress is perceived through SWI/SNF2 and its interaction with *PCNA2*. CKs are perceived through the activation of *LOG10* to the synthesis of dihydrofolate reductase. Key proteins were found during mitosis, such as those of the cytoskeleton, microtubules, tubulins, motor proteins, and chromosome stability, as well as the proteins of the synthesis phase. Like PCNA2, RNR1, TSO2, THY-1, SWI/SNF2.

##### Histone Binding Proteins

Three histone proteins are present in the CC cluster; histone H2A (PGS003DMT40009751), histone H2A.1 (PGS0003DMT40003677), and histone H2AXB (PGSC0003DMT00034958). 

Chromatin structure is essential for normal cell cycle progression, chromosome segregation, and centromere function [96].

Loss of function mutations in *H2A* cause an increase in ploidy phenotype, and an increase rate of chromosome loss and defects in completing the G2/M phase of the cell cycle. Altered centromeric chromatin structure and mutations in genes encoding kinetochore components revealed the role for *H2A* in the proper centromere-kinetochore function [97].

Histones also regulate a DNA damage response by epigenetic modifications. The expression of a loss of function mutant of *H2AX* Ser139Ala, that partially disrupts the phosphorylation site, suppressed the normal G2/M cell cycle arrest following DNA damage by ionizing radiation. The G2/M cell cycle arrest by DNA damage is promoted by checkpoints such as *BRCA1* [98]. Taken together, *H2AXB* (PGSC0003DMT00034958) regulates the DNA damage response via *BRCA1* (PGSC0003DMT400023253) to arrest mitotic cell cycle and ensure an appropriate cell division during MTs development.

These three proteins interact with BRCA1, a heterodimeric ubiquitin E3 ligase, required for the accumulation of ubiquitin conjugates at sites of DNA damage and for silencing at DNA satellite repeat regions. BRCA1 interact with E3 ubiquitin-protein ligase ORTHRUS 2 (PGSC000DMT400084297) and PCNA2.

Histone chaperone ASF1 (PGSC0003DMT400078369) interacts with PCNA2 with a combined score of 0.970 and with THY-1 with a score (0.860). ASF1 play an important role in chromatin replication, maintenance of genome integrity, and cell proliferation during plant development. Histones prevent DNA from becoming tanged, protect it from DNA damage, and play important roles in gene regulation and DNA replication.

MTs development under high content of sucrose/gelrite, 2iP and darkness is activated by PEBP members (StSP6A, StSP5G, StSP3D, CEN1/BFT, ATC) and the TF FD. PEBP members interact with RPs to modulate CC regulation (DNA synthesis and mitosis) (Figure 14).

##### 3.4. Validation of Transcriptomic-Wide Analysis by RT-qPCR

To validate the results of the transcriptomic-wide analysis and the conclusions reached, we performed an analysis by RT-qPCR of 21 genes interacting within the PPI network.

Figure 13 shows the results of validation by RT-qPCR. For all cases, the correct validation is observed as they are up-regulated both in the transcriptome and in the relative quantification.

## 4. Materials and Methods

### 4.1. Plant Material 

#### 4.1.1. Potato Shoot Micropropagation 

Potato cv. Alpha plantlets were propagated by shoot proliferation under in vitro conditions in MS medium [99], supplemented with sucrose 30 g/L (CAT 57-50-1 Sigma-Aldrich, St. Louis, MO, USA), activated charcoal 3 g/L (CAT: 242276 Sigma-Aldrich, St. Louis, MO, USA), pH 5.8 and solidified with gelrite 3 g/L (GELZAN CAT. G1910 Sigma-Aldrich, St. Louis, MO, USA). Shoots were incubated at 25/17 °C under fluorescent light at 25 μmol/m^2^/s of irradiance. Stolon explants derived from propagated shoots were used for MTs induction. 

#### 4.1.2. Potato MTs Induction 

Potato MTs induction protocol was made according to Herrera-Isidron et al. [10]. Stolon explants were cultured in flasks containing MR8-G6-2iP medium: MS medium, supplemented with 2iP 10 mg/L, 8% sucrose, 6 g/L gelrite, 3 g/L activated charcoal, pH 5.8, osmotic potential of 2.02 mPa. As control, stolon explants were cultured in MR1-G3-2iP: MS medium supplemented with 10 mg/2iP, 1% sucrose, 3 g/L gelrite, 3 g/L activated charcoal, pH 5.8, osmotic potential of −0.88 Mpa.). Flasks were sealed with plastic and incubated in dark at 25/17 °C for 15 days of incubation. 

#### 4.1.3. Transcriptome Sequencing and Assembly

To isolate RNA from MTs, tissue samples were collected after 15 days of incubation; at this time, MTs initiate to develop [10].

Total RNA was isolated by using Trizol (Invitrogen, Carlsbad, CA, USA), RNA concentration was measured by its absorbance at 260 nm, ratio 260 nm/280 nm was assessed, and its integrity confirmed by electrophoresis in agarose 1% (*w*/*v*) gels. 

Samples of cDNA were amplified by PCR using SYBRTM Green (ThermoFisher CAT: 4312704, Waltham, MA, USA) in Real-Time PCR Systems (CFX96 BioRad, Herules, CA, USA). 

Final product cDNA samples for sequencing were sent to GENEWIZ, Plainfield, NJ, USA. Illumina HiSeq 2500 (Illumina, San Diego, CA, USA) was used for sequencing. Sequenced reads were tested for quality using the FastQC software package (http://www.bioinformatics.babraham.ac.uk/projects/fastqc/) and preprocessed to remove sequence adapters and low-quality bases using the software Trimmomatics [100].

#### 4.1.4. Analysis of DEG

Adapter removal was performed using the Trimmomatic v0.3.6 program [100]. RNA-seq reads were aligned to the *Solanum tuberosum* reference genome available in Phytozome v12.1. (https://phytozome.jgi.doe.gov/pz/portal.html) with the STAR aligner v.2.5.2b. [101]. In this step, the BAM (Binary Alignment/Map) files were generated. Subsequently, a count and set of transcripts were made using the featureCounts program of the Subread v.1.5.2 package [102]. A quantification and differential analysis of the transcripts was performed using the DESeq2 v1.12.4 program. Finally, an ontology analysis was performed using Blast2GO.

#### 4.1.5. PPI Analysis of Microtuberization 

A gene network with high confidence (0.860) was performed with the STRING database v11.5 [103], based on *S. tuberosum*, homologous genes present in the *S. tuberosum* genome in Sol genomics network [104]. Gene identifier (Id) was made according to the UNIPROT [105], NCBI [106] database. Homologous in *S. tuberosum* greater than 60% in protein sequence with *A. thaliana* were considered. Oligonucleotides were designed to qPCR (2−∆∆CT method analysis) [107] gene expression or transcriptional analysis (Appendix A). 

## 5. Conclusions

A PPI network of up-regulated TFs revealed that at least six TFs–MYB43, TSF, bZIP27, bZIP43, HAT4, WOX9–may be involved during MTs development.Two fundamental biological process essential for life and highly conserved through organisms were found interacting tightly: RPs comprising 29 and CC 117 proteins.PEBP members interact with RPs and CC process to activate MTs development under high content of sucrose and gelrite, 2iP under darkness.Further experiments by yeast two-hybrid screening approach of genome edited up- and down-regulated PEBP members with RPL11 are required to demonstrate our model of MTs development under darkness.

## Figures and Tables

**Figure 1 ijms-23-13835-f001:**
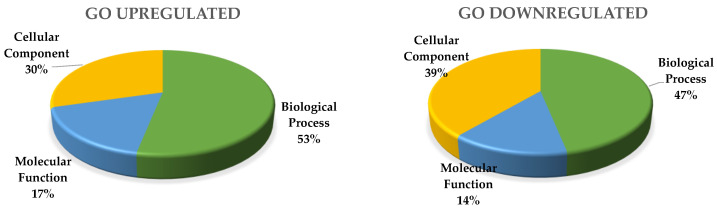
Biological process regulated during MTs development of potato induced in dark.

**Figure 2 ijms-23-13835-f002:**
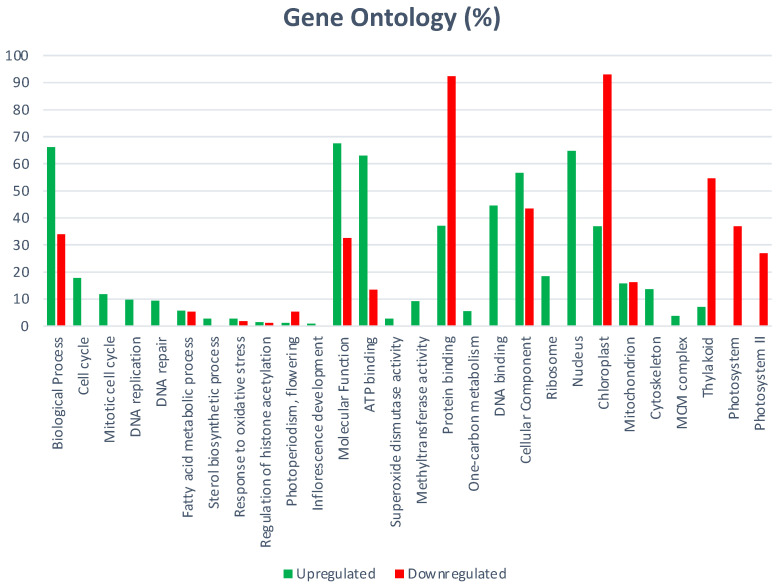
Cellular components regulated during MTs development of potato induced in the dark.

**Figure 3 ijms-23-13835-f003:**
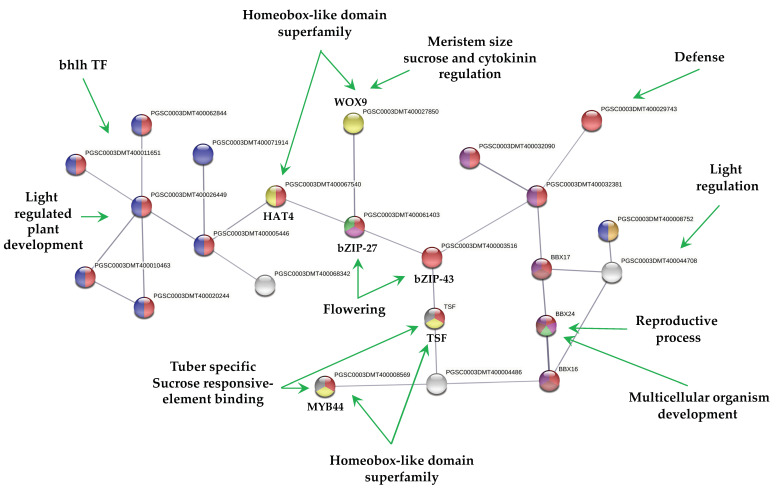
PPI network of up-regulated TFs derived using STRING database from transcriptomic-wide analysis of potato *S. tuberosum* with medium confidence (0.400). The figure represents a full network; the edges indicate both functional and physical protein associations.

**Figure 4 ijms-23-13835-f004:**
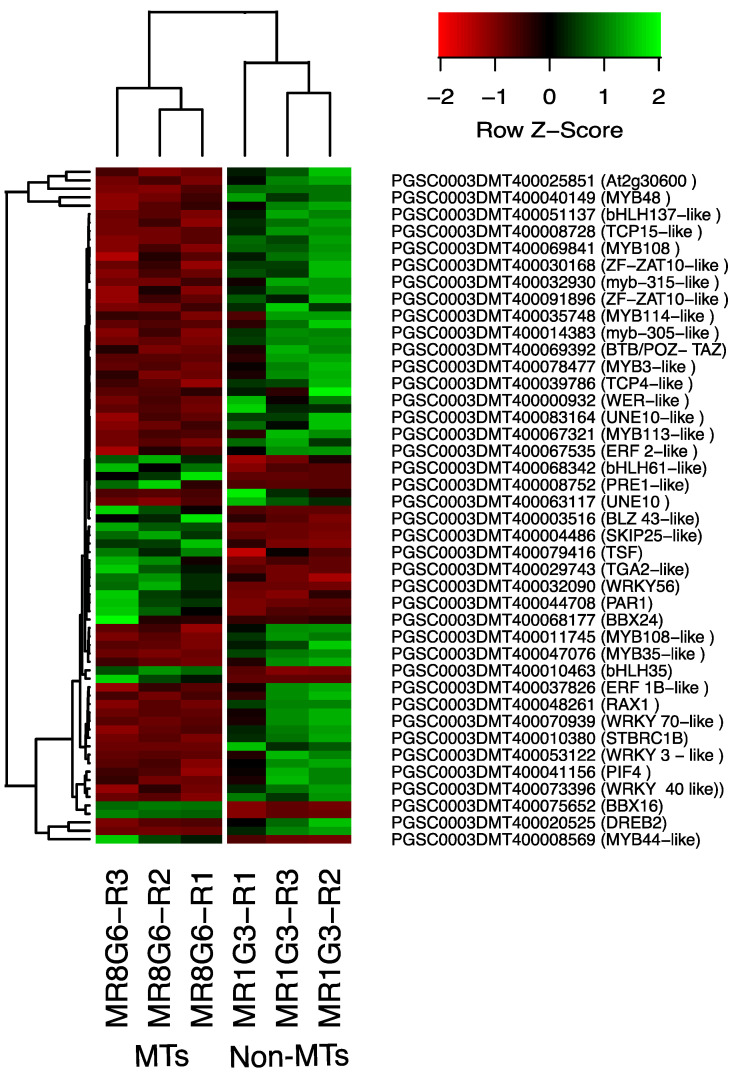
Hierarchical clustering analyses (HCA) and heat map of up- and down-regulated TFs during MTs development under darks conditions, levels of up-regulation are present in Log2.

**Figure 5 ijms-23-13835-f005:**
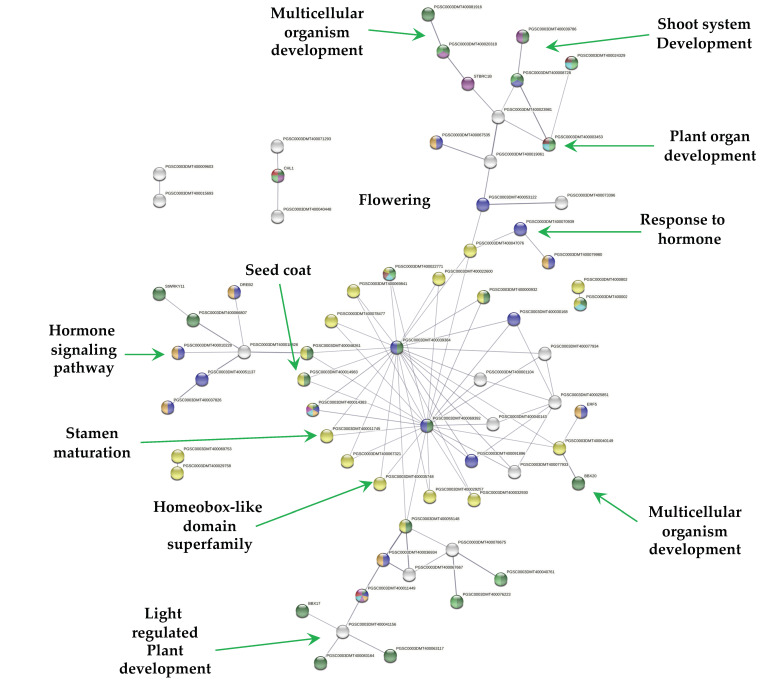
The PPI network of down-regulated TFs derived using the STRING database from the transcriptomic-wide analysis of potato *S. tuberosum* with medium confidence (0.400). The figure represents a full network, the edges indicate both functional and physical protein associations.

**Figure 6 ijms-23-13835-f006:**
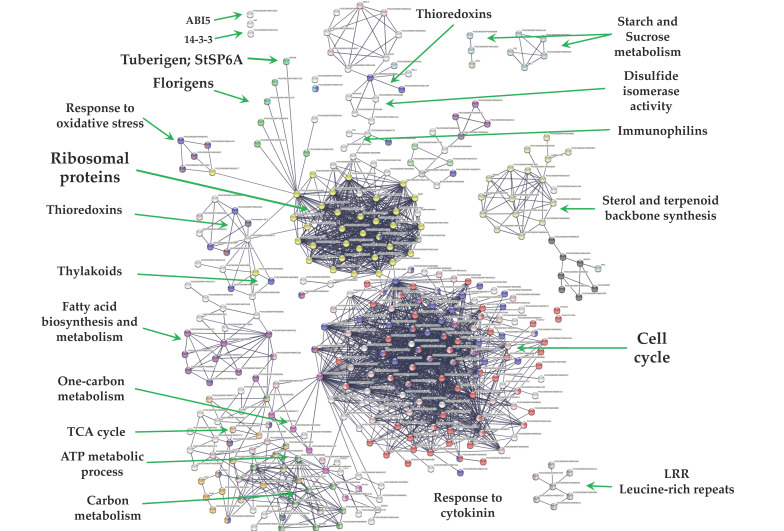
PPI network of up-regulated genes derived from the STRING database of potato *S. tuberosum* from the transcriptomic-wide analysis with high confidence (0.860). Clusters are highlighted with the name of the function. The figure represents a full network, the edges indicate both functional and physical protein associations.

**Figure 7 ijms-23-13835-f007:**
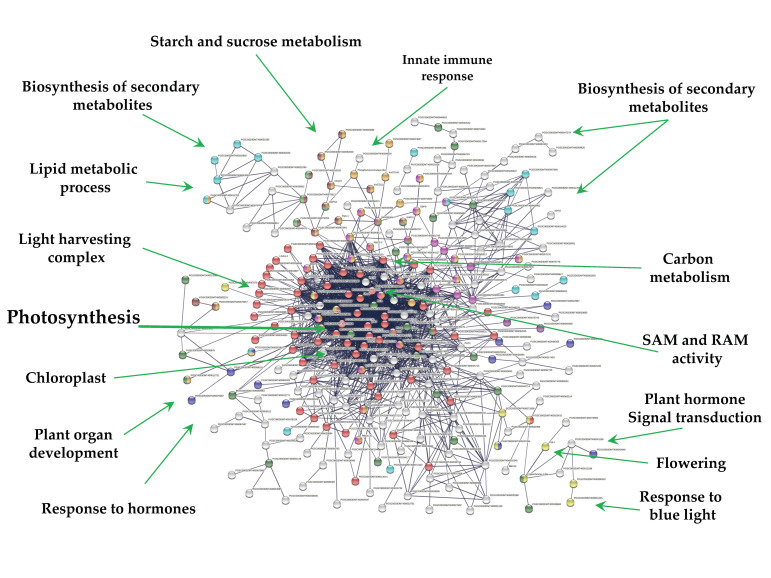
PPI network of down-regulated genes derived from the STRING database of potato *S. tuberosum* from the transcriptomic-wide analysis with high confidence (0.700). Cluster is highlighted with the name of the function. The figure represents a full network, the edges indicate both functional and physical protein associations.

**Figure 8 ijms-23-13835-f008:**
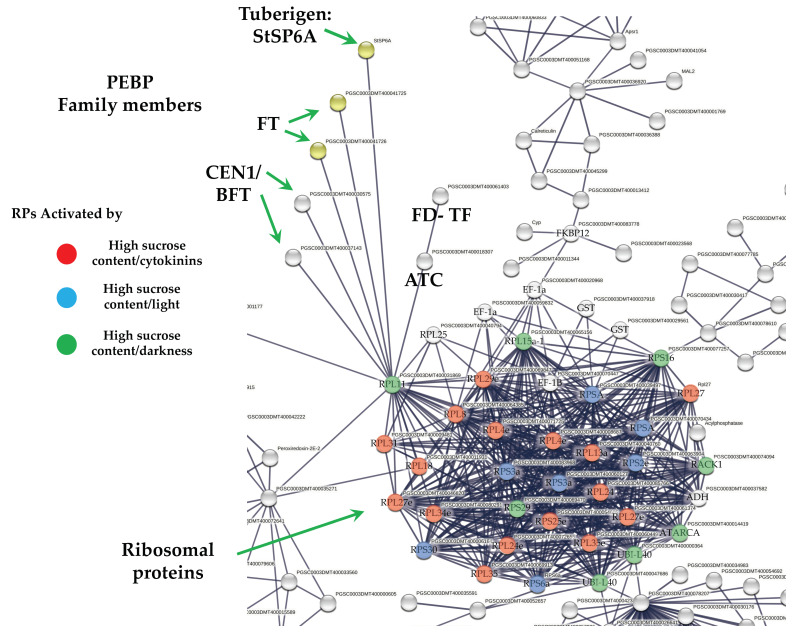
PPI network of up-regulated PEBP family members and RPs cluster of potato S. tuberosum from the transcriptomic-wide analysis with high confidence (0.860). Selected proteins are highlighted with the name of the gene. RPs highlighted in red, blue, and green correspond to activated proteins by sucrose and CKs. The list of RPs and levels of gene regulation are shown in the heat map (Figure 9).

**Figure 9 ijms-23-13835-f009:**
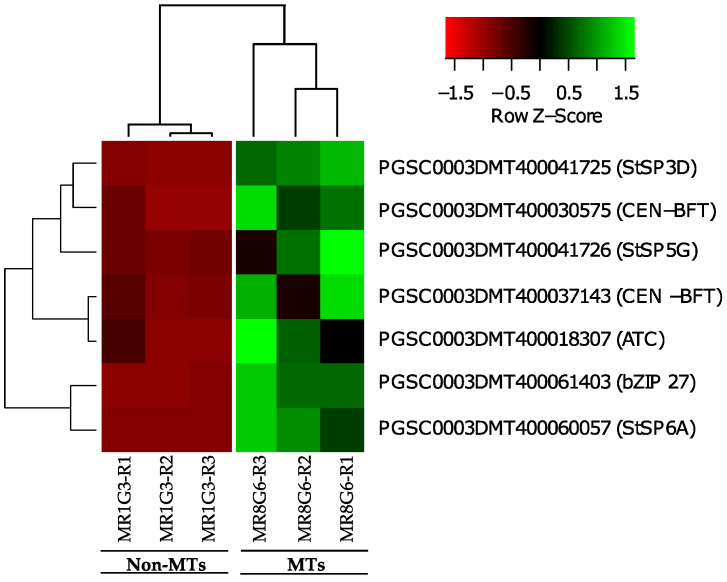
Hierarchical clustering analyses (HCA) and heat map of PEBP family members during MTs development under darks conditions; levels of up-regulation are present in Log2.

**Figure 10 ijms-23-13835-f010:**
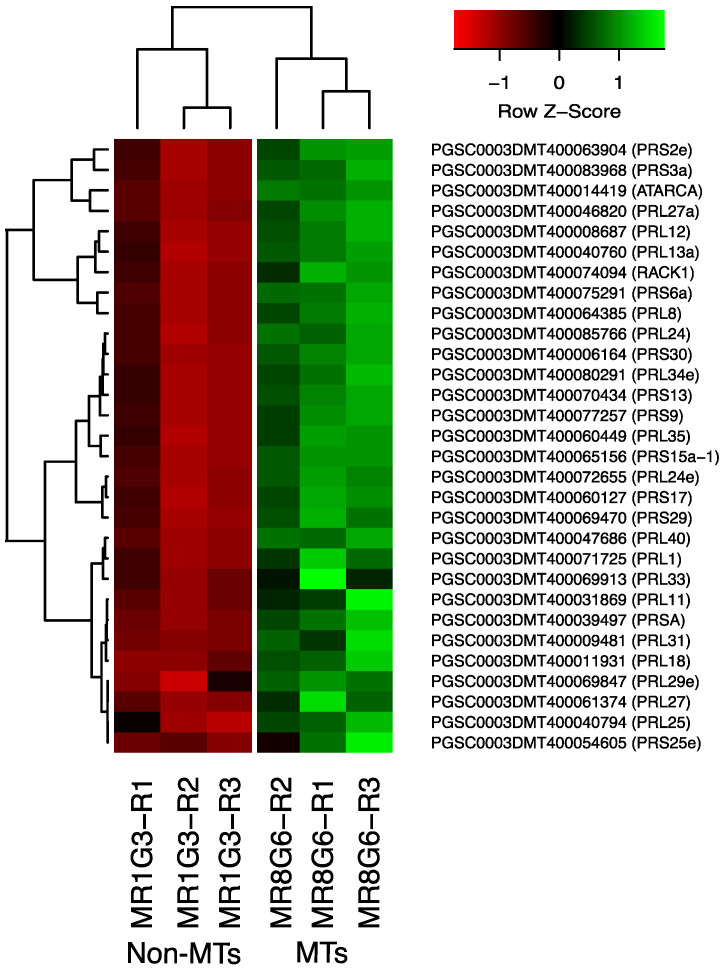
Hierarchical clustering analyses (HCA) and heat map of RPs during MTs development under darkness conditions from levels of up-regulation are present in Log2.

**Figure 11 ijms-23-13835-f011:**
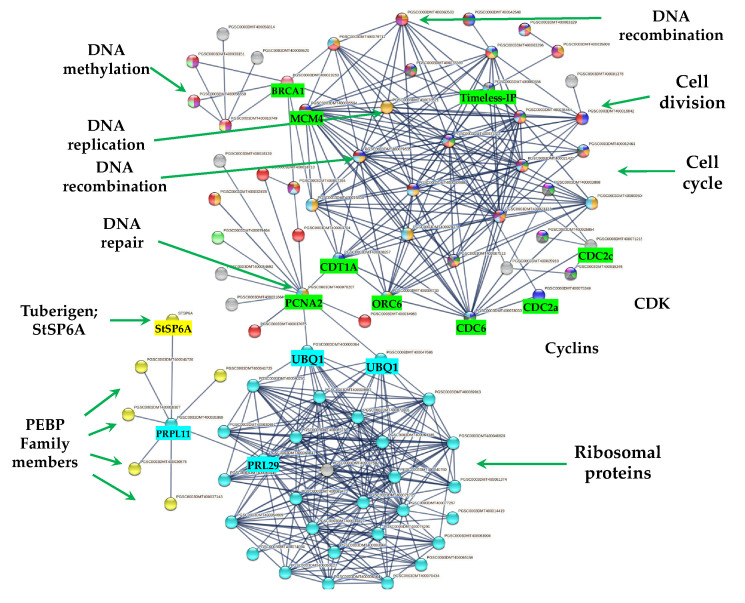
PPI physical network of up-regulated transcripts genes of potato S. tuberosum from the transcriptomic-wide analysis with high confidence (0.860). Selected proteins are highlighted with the name of the gene.

**Figure 12 ijms-23-13835-f012:**
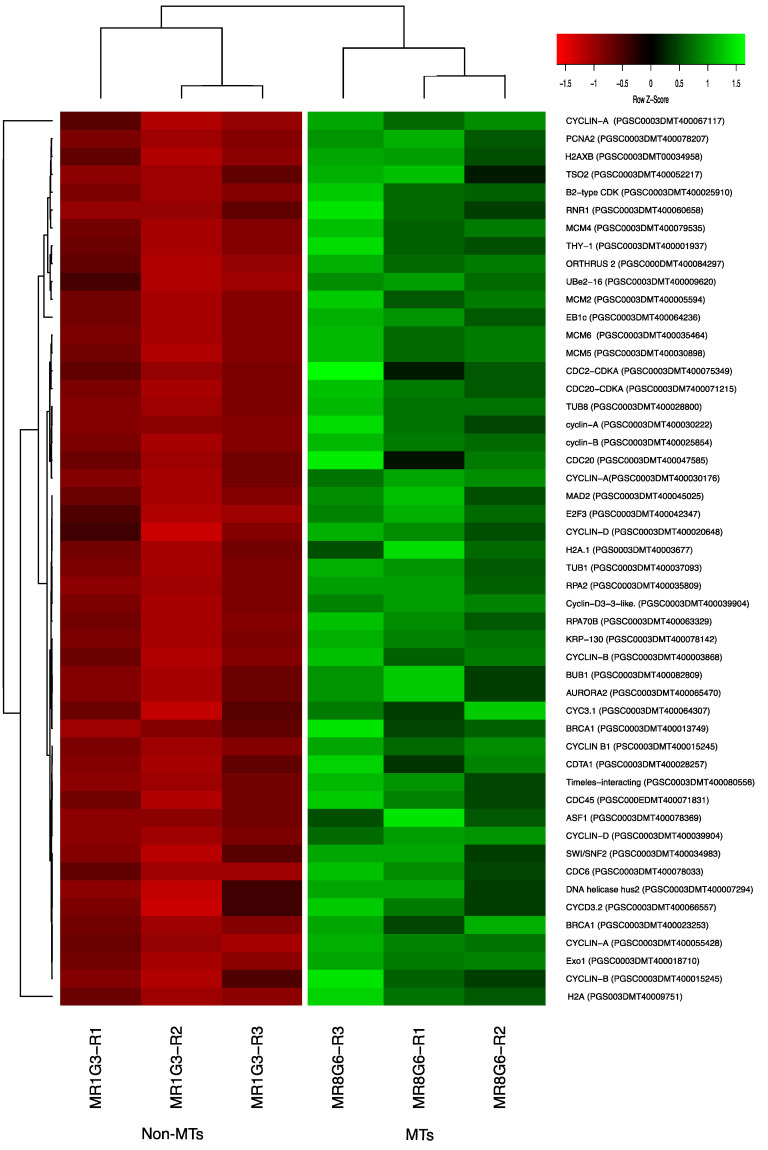
Heat map of CC regulation proteins during MTs development under dark conditions; levels of up-regulation are present in Log2.

**Figure 13 ijms-23-13835-f013:**
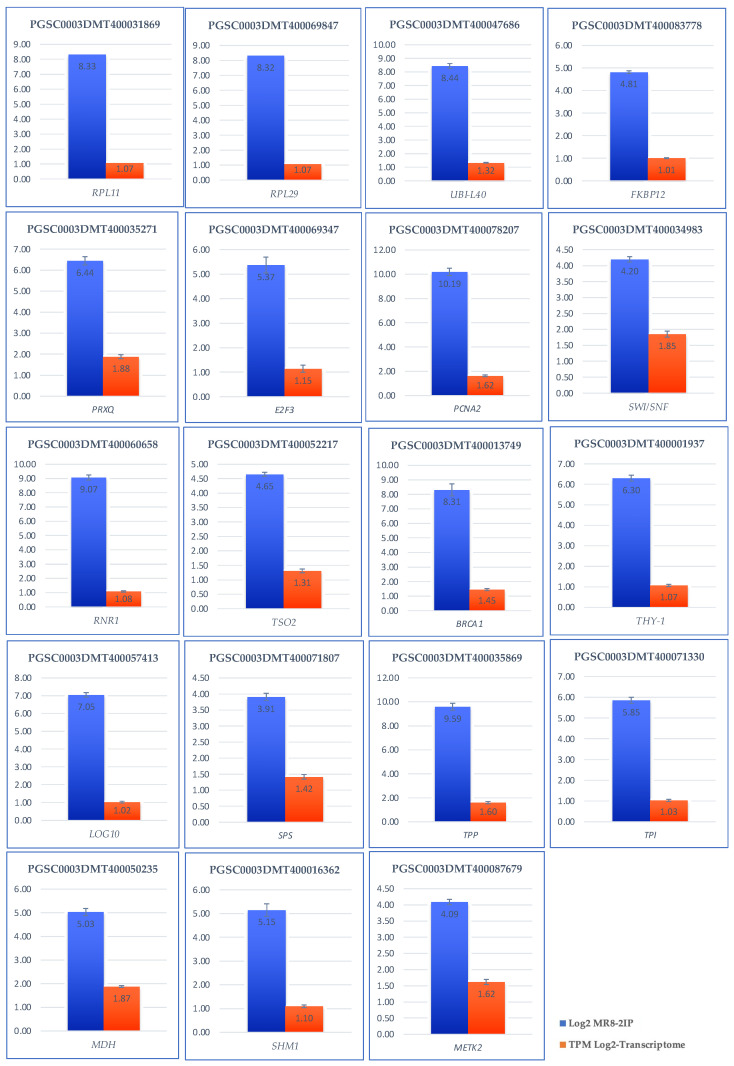
Validation of the transcriptomic-wide analysis by quantitative reverse transcription PCR (qRT-PCR) of 19 DEG up-regulated genes involved in RPs, PEBP, CC, carbon metabolism of potato *S. tuberosum*. Blue columns correspond to absolute gene expression derived from the genome-wide analysis. The orange bars represent the expression of the transcriptome expressing the number of transcripts per million.

**Figure 14 ijms-23-13835-f014:**
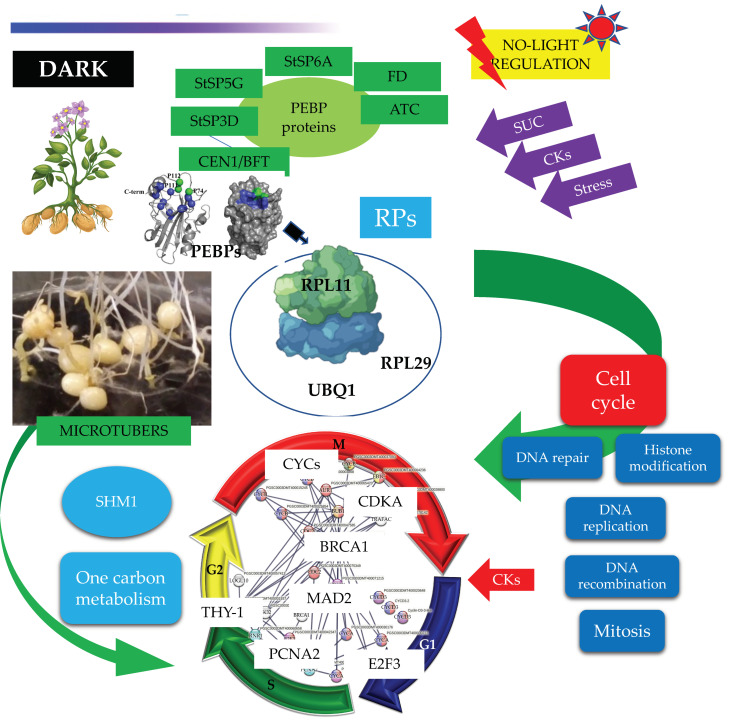
PPI network of up-regulated CC proteins involved in mitosis (M), S, G2, G1 phase of potato S. tuberosum from transcriptomic-wide analysis with high confidence (0.860).

**Table 1 ijms-23-13835-t001:** Mapping statistics for quality filtered reads generated for lines with MTs and non-MTs induction.

	MR8G62IP (MTs)	MR1G32IP (non-MTs)
Read -1	58,784,802.00	71,588,702.00
Read -2	59,487,917.00	73,230,643.00
Read -3	59,185,434.00	75,556,776.00
Yield (Mbases)	177,458,153.00	220,376,121.00

Q40: Error probability 0.0001 (1 in 10.000 base calling).

## Data Availability

Not applicable.

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
