# Peer review of "Solanum tuberosum Microtuber Development under Darkness Unveiled through RNAseq Transcriptomic Analysis"

_ijms, 2022, doi:10.3390/ijms232213835_

Round 1
Reviewer 1 Report
The manuscript entitled “Solanum tuberosum Microtuber development under darkness unveiled through RNAseq Transcriptomic analysis” by Valencia-Lozano et al. revealed the underlying molecular mechanisms involved in microtuber formation, where the microtuber development is regulated by interaction of PEBP members with the fundamental processes for life, ribosomal proteins, interacting with molecular factors that activate the development of microtubers. Therefore, their outcomes are strongly relevant with the readership of International Journal of Molecular Sciences. The revised manuscript can be considered for acceptance after the following minor issues are addressed.
1. The Abstract need to be summarized in one paragraph, and to more concisely summarize the research highlights.
2. The Conclusion need to be rewritten to conclude the work deficiencies and future work prospects of this paper.
3. The introduction should focus more on the existing work and highlight the research innovation and characteristics of this paper.
4. The paragraphs of “Plant biotechnology techniques have been applied to produce potato tubers in vitro called microtubers (MTs), minitubers and vitrotubers”, “MTs process occurred when axillary buds are cultured in medium containing...” and “MTs has advantages in terms of storage, transport, and cultivation,...” in the introduction can be summarized in a whole, for the same statement of the background of microtubers.
5. The format of some references exist problems. The whole references are too old and need to be updated.
Author Response
The Abstract need to be summarized in one paragraph, and to more concisely summarize the research highlights.
Corrected
Potato microtuber development through in vitro techniques are ideal propagules for producing high quality potato plants. Microtuber formation is influenced by several factors, i.e., photoperiod, sucrose, hormones and osmotic stress. We have previously developed a protocol of microtuber induction in medium with sucrose (8%w/v), gelrite (6g/L), and 2iP as cytokinin under darkness. To understand the molecular mechanisms involved, we performed a transcriptome-wide analysis. Here we show that 1715 up and 1624 down-regulated genes were involved in this biological process. Through the protein-protein interaction network analyses performed in the STRING database (v11.5, www.string-db.org) we found 299 genes tightly associated in 14 clusters. Two major clusters of up-regulated proteins, fundamental for life growth and development were found: 29 ribosomal interacting with 6 PEBP family members and 117 cell cycle proteins. The PPI network of up-regulated TFs revealed that at least 6 TFs MYB43, TSF, bZIP27, bZIP43, HAT4, WOX9 may be involved during MTs development. The PPI network of down-regulated genes revealed a cluster of 83 proteins involved in light and photosynthesis, 110 in response to hormone, 74 in hormone mediate signaling pathway and 22 related to aging.
Keywords: Transcriptome-wide analysis; Microtubers; Potato; Solanum tuberosum; Darkness; Cell cycle; Ribosomal proteins; PEBP family genes; Cytokinin.
The Conclusion need to be rewritten to conclude the work deficiencies and future work prospects of this paper.
Corrected
5. Conclusions
1.- A PPI network of up-regulated TFs revealed that at least 6 TFs MYB43, TSF, bZIP27, bZIP43, HAT4, WOX9 may be involved during MTs development.
2.- Two fundamental biological process essential for growth and development and highly conserved through organisms were found interacting tightly: RPs comprising 29 and CC 117 proteins.
3.- PEBP members interact with RPs and CC process to activate MTs development under high content of sucrose and gelrite, 2iP under darkness.
4.- Further experiments by yeast two-hybrid screening approach of genome edited up and down-regulated PEBP members with RPL11 are required to demonstrate our model of MTs development under darkness.
The introduction should focus more on the existing work and highlight the research innovation and characteristics of this paper.
The introduction was changed, 21 references from the molecular mechanisms of potato tuberization were eliminated and additional information about the PPI network of upregulated TFs were added, as well as our main findings. See the corrected manuscript.
- The paragraphs of“Plant biotechnology techniques have been applied to produce potato tubers in vitro called microtubers (MTs), minitubers and vitrotubers”, “MTs process occurred when axillary buds are cultured in medium containing...” and “MTs has advantages in terms of storage, transport, and cultivation,...” in the introduction can be summarized in a whole, for the same statement of the background of microtubers.
Corrected
Plant biotechnology techniques have been applied to produce potato tubers in vitro called microtubers (MTs). MTs process occurred when axillary buds are cultured in medium containing high content of sucrose, hormones and different light quality/darkness. MTs has advantages in terms of storage, transport, and cultivation, due to their reduced size and weight, requiring less space compared to seedlings, with higher multiplication rate producing seed potato faster and cheaper than other methods [For review; 31-34].
- The format of some references exist problems. The whole referencesare too old and need to be updated.
21 references were eliminated, some old were included due to its importance in the discussion.

Reviewer 2 Report
This descriptive manuscript provides an extensive source of documented genetic mechanistic data and should be published for it's body of knowledge pertaining to potato gene action and micro-tuberization. Fig.10 provides an excellent process summary.
L22: Potato microtubers, produced by in vitro techniques can be a source of high- quality potato propagules. ( Microtubers are not seeds, per se, thus a rewording is necessary.
L29: It was revealed that (not necessary in an abstract, start with number of genes)
L35: As it was expected,
L37: hormones, …pathway, and 22….
L39: life growth, …
L41: CC clusters that interact…
L57: The Potato tubers…
L78: MTs process occurred…
L100: metabolism and hormone…
L134: flowering, and plant organ development.
L374: darkness conditions; from…
L448: transcripts genes…
L484: The orange bars represent..
L529: a
L567: clusters…
L586: A few…
L588: swolloing swelling…
L594: karakins, as well as repression…
L748: cell growth to correctly develop an organ..
L952: life …
Author Response
Answers for Reviewer 2
This descriptive manuscript provides an extensive source of documented genetic mechanistic data and should be published for it's body of knowledge pertaining to potato gene action and micro-tuberization. Fig.10 provides an excellent process summary.
L22: Potato microtubers, produced by in vitro techniques can be a source of high- quality potato propagules. ( Microtubers are not seeds, per se, thus a rewording is necessary.
Dear reviewer 2. We have made all corrections.
Corrected, eliminated potato seeds
L29: It was revealed that (not necessary in an abstract, start with number of genes)
Corrected, It was revealed that,,, was eliminated.
L35: As it was expected,
Corrected, eliminated As it was expected
L37: hormones, …pathway, and 22….
Corrected, and was added
L39: life growth, …
Corrected, growth was added
L41: CC clusters that interact…
Corrected, that was added
L57: The Potato tubers…
Not applicable
L78: MTs process occurred…
Corrected
L100: metabolism and hormone…
Corrected
L134: flowering, and plant organ development.
Corrected
L374: darkness conditions; from…
Corrected
L448: transcripts genes…
Corrected
L484: The orange bars represent..
Corrected
L529: a
Corrected
L567: clusters…
Corrected
L586: A few…
Corrected
L588: swolloing swelling…
Corrected
L594: karakins, as well as repression…
Corrected
L748: cell growth to correctly develop an organ..
Corrected
L952: life …
Corrected
